# Lumpy Skin Disease Virus Genome Sequence Analysis: Putative Spatio-Temporal Epidemiology, Single Gene versus Whole Genome Phylogeny and Genomic Evolution

**DOI:** 10.3390/v15071471

**Published:** 2023-06-28

**Authors:** Floris C. Breman, Andy Haegeman, Nina Krešić, Wannes Philips, Nick De Regge

**Affiliations:** Sciensano, Unit Exotic and Vector Borne Diseases (ExoVec), Groesselenberg 99, B-2800 Ukkel, Belgium; andy.haegeman@sciensano.be (A.H.); nina.kresic@sciensano.be (N.K.); wannes.philips@sciensano.be (W.P.); nick.deregge@sciensano.be (N.D.R.)

**Keywords:** lumpy skin disease, genome, phylogeny, evolution

## Abstract

*Lumpy Skin Disease virus* is a poxvirus from the genus *Capripox* that mainly affects bovines and it causes severe economic losses to livestock holders. The *Lumpy Skin Disease virus* is currently dispersing in Asia, but little is known about detailed phylogenetic relations between the strains and genome evolution. We reconstructed a whole-genome-sequence (WGS)-based phylogeny and compared it with single-gene-based phylogenies. To study population and spatiotemporal patterns in greater detail, we reconstructed networks. We determined that there are strains from multiple clades within the previously defined cluster 1.2 that correspond with recorded outbreaks across Eurasia and South Asia (Indian subcontinent), while strains from cluster 2.5 spread in Southeast Asia. We concluded that using only a single gene (cheap, fast and easy to routinely use) for sequencing lacks phylogenetic and spatiotemporal resolution and we recommend to create at least one WGS whenever possible. We also found that there are three gene regions, highly variable, across the genome of LSDV. These gene regions are located in the 5′ and 3′ flanking regions of the LSDV genome and they encode genes that are involved in immune evasion strategies of the virus. These may provide a starting point to further investigate the evolution of the virus.

## 1. Introduction

Lumpy Skin Disease (LSD) is caused by *Lumpy Skin Disease virus* (LSDV). Lumpy Skin Disease affects mainly bovines, whereas the other members of the genus *Capripox*, including *Sheeppox* (SPPV) and *Goatpox* (GTPV), mainly infect sheep (*Ovis aries* s.l.) and goats (*Capra hircus* s.l.), respectively. LSDV is mainly transmitted by vectors [1] while other transmission routes play a minor role in transmission [2,3,4,5,6]. Often observed clinical signs are the occurrence of lachrymation and nasal discharge, fever, reduced appetite, reduced milk production and the occurrence of nodules on the skin, mucous membranes and the organs. The nodules are diagnostic for LSD,, but the severity is dependent on the susceptibility of the host and the virulence of the strain [7]. The nodules may occur all over the body and may become necrotic, thus providing a pathway for secondary infections. Importantly, a significant part of the infected animals remains subclinical [8], but these animals may nevertheless contribute to transmission of the disease [9]. The morbidity rate ranges from 5 to 45% [10,11]. Several homologous vaccines were created based on the Neethling LSDV strain; they are effective in providing protection against LSDV infections in cattle [12]. Another group of vaccines are those based on the LSDV KSGP strain which was originally created for the vaccination of sheep and goats against *Sheep*- and *Goatpox*, [13], respectively, as it was long-believed to be a sheep–goat pox strain [13,14,15].

The LSDV is classified into the family *Poxviridae*, subfamily *Chordopoxvirinae*, genus *Capripox*. It consists of double-stranded DNA and its genome is ~150 kb in length. Its type strain (GenBank accession code NC_003027) is based on a sample collected in 1958 in the republic of South Africa and sequenced by [16]. The classical wild-type field strains of LSDV are divided into two major clades (1.1 and 1.2, respectively) [17,18]. There appears to be no clinical difference between the strains from these two clades, but they are different from a genomic perspective.

Strains from Clade 1.2 were introduced from the Middle East to Europe and Eurasia in the Balkans [19], Russia and Kazakhstan in 2014–2015 [3]. Large-scale vaccination campaigns were initiated following this outbreak. One of the imported and used vaccines turned out to be insufficiently quality controlled [12] and contained, among others, recombinant LSDV strains that were probably released in the field during the vaccination campaign in Kazakhstan [20]. This resulted in the introduction of an entirely new field strain which quickly spread throughout Southeast Asia [21,22]. This vaccine-derived recombinant strain is characterized by combined genetic sequences from known vaccine strains from Clades 1.1 (Neethling strain) and 1.2 (KSGP strain) and possibly even *Goatpox* sequences [20,23].

During the last decade, and certainly after the LSDV incursion in Europe, an increasing number of historical and emerging LSDV strains from outbreaks worldwide were sequenced. The strain characterization/classification often relies on the sequence of one or a limited number of genomic regions (e.g., GPCR, RPO30), which leads to a low resolution in-or poorly resolution of phylogenetic trees. This in turn prevents a more detailed analysis of patterns of phylogenetic relationships. The recent discovery and broad implementation of third-generation sequencing methods, however, has resulted in more whole genome sequences (WGS) of LSDV strains to become available.

*Poxviridae* are, as a rule, slow-evolving viruses when compared to other viruses, and *Capripox* are no exception to this [24]. The substitution rate in the extended conserved central region of the *Variola* virus (genus *Orthopox*) was estimated at ~0.9–1.2 × 10^−6^ substitutions/site/year [25]. This was estimated to be 1.0 × 10^−5^ substitutions/site/year for the *Myxoma* virus genome [26] and recently 7.4 × 10^−6^ substitutions/site/year for the LSDV genome [18]. Genomic variation (sequence variation, rearrangements, gene gain/loss) in poxviruses seems to occur much more frequently at the flanks of the genome and in genes involved in immune-evasion of the hosts’ immune system, whereas genes located in the central part of the genome (the ‘core pox genome’) seem to be more conserved [27,28,29].

In this study, we took advantage of all partial- and whole-genome sequences that were deposited in public repositories by February 2023 to gain insight in several epidemiological and fundamental research questions related to LSDV. First, a phylogenic and network analysis of whole-genome sequences was used to propose a putative spatio-temporal spread of LSDV. Second, whole-genome sequences were compared to single-gene-based sequences for studying *Capripox* evolution and correct phylogenetic placement of newly detected strains. Third, we analyzed whether classical and recent recombinant field LSDV strains are under similar selective pressures and whether similar genes as in other poxviruses evolve faster. We aimed to identify these highly variable regions as they could serve as the focus of future research.

## 2. Materials and Methods

### 2.1. WGS Data Used

All WGS for LSDV that were available on 01-02-2023 were collected, in addition to two outgroup sequences for *Sheeppox* and *Goatpox* (GenBank accession numbers NC004002 and NC_004003, respectively). This dataset also included vaccine sequences. Sequences were excluded/omitted when (1) no GenBank number was available; (2) they were human-made mutants; (3) origin could not be determined (either literature, geographic origin or isolate) or (4) when the sequence contained more than >1% sequence ambiguities (e.g., MT007951). Sequences were trimmed at the beginning (323 bp) and end (206 bp) due to very little overlap between sequences at these terminal locations as well as a high number of ambiguous bases at these sites in some sequences. We constructed two datasets. The first dataset contained only sequences collected before 2017 (labelled hereafter as the ‘<2017′ dataset) corresponding to the situation before circulation of the vaccine-derived recombinant field strains. The second dataset contained all currently available sequences, including those from strains collected after 2017 (labelled hereafter as the ‘>2017′ dataset). A full overview of the data and references used can be found in Supplementary Material Appendix A. In the results, we refer to a clade when a grouping of sequences is based on an analysis with specific evolutionary models underpinning this (be they posterior probabilities based on Bayesian statistics, likelihood- or distance-based models) and statistical support by these models for the nodes at the basis of each clade is found. In all other cases, a grouping of sequences is named a ‘cluster’.

### 2.2. Single Gene Datasets

From the >2017 dataset, we extracted exons of five genes which are frequently used for LSDV diagnostics and/or phylogeny reconstruction: (1) RNA polymerase 30 kDa subunit (rpo30); (2) G protein-coupled receptor (GPCR); (3) DNA Polymerase (DNA_Pol); (4) the LSDV envelope protein (P32), and the (5) RNA polymerase 132 kDa subunit (rpo132). These were analyzed using the same phylogenetic analysis as for the WGS datasets, as described below. The models used for the phylogeny reconstruction are described in Supplementary Material Appendix A.

### 2.3. Phylogeny Reconstruction

Phylogenetic analyses used to visualize a hypothesis of evolutionary relationships of our datasets were performed under Maximum Likelihood (ML) criteria using IQ-TREE multicore version 1.6.12 for Windows run locally (http://iqtree.cibiv.univie.ac.at/ accessed on 22 February 2023) [30,31,32]. Sequences that are exactly alike are discarded by IQ-TREE during the analysis and later added to the phylogeny to prevent the occurrence of inflated support values for nodes during bootstrapping. We used model selection (ModelFinder [33]) to find the optimal model. Iqtree uses three criteria for model selection: (1) the Akaike information criterion (AIC); (2) the corrected Akaike information criterion (cAIC); and (3) the Bayesian information criterion (BIC). In case of conflict, Iqtree employs the model selected by the BIC. We used bootstrapping (UFBoot) to generate 10,000 trees. We used the Shimodaira–Hasegawa-like approximate Likelihood ratio testRT (SH-aLRT) [34], the approximate Bayes test and bootstrapping to evaluate node support. We further employed nearest neighbor interchange (NNI) search to initialize the candidate set and increased this to 100 (as opposed to 20 under default settings), and during the likelihood search, we kept the ten best trees during each step rather than the default of five. We also stored the trees with branch lengths for evaluation in the reconstruction of consensus networks.

For the <2017 dataset, the TVM + F + R2 model was selected. All three selection criteria employed by Iqtree were in agreement. The model is a ‘transversion model’ where AG = CT and unequal base frequencies are considered (TVM). The base frequencies were determined empirically (‘F’) and a free rate heterogeneity (‘R2′ with two categories) was assumed. For the >2017 dataset, the K3Pu + F + R4 model was selected according to the BIC. The AIC and cAIC model optimization arrived at the TIM + F + R4 model, but given the small differences between these models, we opted to use the slightly more complex model selected under the BIC. The K3P model is a model assuming three modes of nucleotide substitutions. Just as for the <2017 dataset, the base frequencies were determined empirically (‘F’) and a free rate heterogeneity (‘R4′) was assumed. For a detailed overview of the models available and the selection please, see [35] and (http://www.iqtree.org/doc/Substitution-Models accessed on 22 February 2023) for more recent additions.

Clades in phylogeny are considered supported when the three employed methods of node evaluation, Shimodaira–Hasegawa-like approximate Likelihood ratio testRT (SH-aLRT) [34], the approximate Bayes test (pp) and bootstrapping (BS-s), yield statistical support of values >75/>97/75, respectively.

### 2.4. Network Reconstructions

In order to visualize the lower branches on the trees as well as possible conflicts between the trees reconstructed using the bootstrap analysis, we took the 10,000 trees generated by the ML analysis and constructed a consensus network for each of the WGS-based datasets (i.e., the dataset with and without the recombinant strains). The analysis was performed using SplitsTree5 5.0.0_alpha [36,37]. The Splits Network Algorithm method [38] was used (default options) to obtain a splits network using the option to count the edge weights [39]. The threshold for conflict between trees to be reported was set at 10%. We also mapped the total number of variable positions that can be found for each clade that was supported in the phylogeny. While this does not necessarily indicate that these sites are synapomorphic (and therefore they do not convey relevant evolutionary information per se), it nevertheless offers an idea of the level of variation that exists between the sequences in a clade and how different each clade is when compared to the rest of the phylogeny. It also allows to zoom in on possibly closely related strains and identify relevant mutations and allows to determine whether conflicts that occur in the network are caused by possible recombinants or by a lack of resolution.

We added smaller decomposed split networks—using SplitsTree5 5.0.0_alpha based on sequence data of two subclades—to analyze the spatio-temporal patterns in greater detail [40,41]. In the decomposed split networks we excluded the vaccine sequences as we wished to clarify the population structure These structures enable us to examine how characters (nucleotide differences) support divisions of taxa without presupposing a tree-like structure. The split decomposition method dissects a given dissimilarity measure as a sum of elementary ‘split’ metrics plus a (small) residue. The identified related groups are susceptible to further interpretation when ‘casted against the available biological information [40]’.

### 2.5. Genomic Substitution Hotspots in LSDV WGS Datasets

We generated pi (π)-plots in DNAsp V6 [42] to create an overview of the variation within the genomes. We performed this for a dataset containing only the classical wild-type strains, a dataset that only contained the recombinant LSDV sequences, and a dataset that combined all these sequences (this set is equal to the >2017 dataset). The π-plots were reconstructed with the inclusion of indel (‘insertion/deletion’) sites using a sliding window of 100 bp and steps of 25 bp along which the sliding window moved. The average π-values + three times the standard deviation and 10 times the average π-values were used as cutoff values to highlight genomic regions with high levels of substitutions. The names of the genes/genomic regions with values above these cutoff values were derived from [16], and corresponding hypothetical protein functions for each gene were described by [17].

## 3. Results and Discussion

### 3.1. WGS-Based Phylogeny Reconstruction and Deduced Putative Spatio-Temporal Spread

The <2017 dataset contained 28 sequences, the >2017 dataset contained 63 sequences. Both datasets contained sequences of 151,696 bp long representing >99% of the entire genome. Relative to the outgroups, the LSDV in-group of the <2017 dataset sequences contained 1992 variable positions. The >2017 dataset contained 2243 variable positions in total.

The phylogeny of the <2017 dataset (Figure 1A) revealed eight clades. These correspond to Clades 1.1 (historical South African) and 1.2 (Pan-African/Eurasian) as described before [17,18,43] as well as to six subclades in each group which are based on sequences collected in close spatio-temporal succession of each other (e.g., MN636838-MN636842). Subclades that can be distinguished here are an East African clade as well as an African/Eurasian clade. 

The phylogeny of the >2017 dataset revealed 19 clades (Figure 1B). Clades 1.1 and 1.2 are recovered again as well as a clade corresponding to the recombinant Clade 2.5. Three of the recombinant sequences (belonging to clusters 2.1 and 2.4) appear as sister branches to Clades 1.1 and 2.5, the other two sequences (belonging to clusters 2.2 and 2.3) branch off before Clade 2.5 and are sisters to both Clades 1.2 and 2.5. Within Clades 1.1 and 1.2, the vaccine-based sequences form separate clades together with their apparent classical wild-type ‘ancestors’ (or ‘source’). Furthermore, five Clade 1.1 samples collected in the Republic of South Africa (RSA) in close spatio-temporal succession form a supported clade. Within Clade 1.2 sequences from India and Bangladesh from 2019 cluster within the East African clade containing older samples from Kenya (1974 and 1958) as well as the KSGP vaccine strain and from the East African/Indian clade (1.2.2). A second subclade in Clade 1.2 comprises all sequences from the European/Kazakhstan outbreak and it also contains the few other African sequences from Nigeria and the RSA/Namibia, forming the African/Eurasian clade (1.2.1). In Clade 1.2.1, there are two more subclades, which comprise South African sequences (Subclade 1.2.1.1) and a clade containing only the sequences from Europe/Kazakhstan (Clade 1.2.1.2). The single sequence from Nigeria (OK318001) is sister to both these subclades and it is basal in Clade 1.2.1. Within Clade 2.5 comprising sequences from recombinant strains, two clades can be distinguished. Both these clades contain sequences from East and Southeast Asia (China, Taiwan, Vietnam, Thailand), but one supported clade also contains Russian sequences (with only 20 variable positions between them). The two earliest collected sequences within this clade represent two samples (OM984486 and MW355944) collected in western China (Xinjiang) in 2019 and these appear, based on the phylogeny, to be the sister accessions to all other sequences generated later in 2020–2022 in East and Southeast Asia as well as in Russia (Figure 2 and Figure 3). As these are difficult to see in this figure, a cladogram of Figure 1B was created and displayed in Appendix A. Sequences belonging to Strains 2.1–2.4 have only be found on single occasions in Russia and were not reported afterwards. This can mean that they no longer circulate because of a selective advantage of Clade 2.5 strains or simply go unnoticed. Given the abundance of sequences recovered from Clade 2.5, the former option is the most likely. 

The WGS-based tree provides more resolution when compared to the single-gene trees, and relations between shallower branches of the phylogenetic tree are sometimes well resolved (e.g., the East African/Indian Clade 1.2.2 and the Eurasian Subclade 1.2.1.2 (Figure 1B and Figure 3). Also, the produced consensus networks provide a good way to visualize and evaluate these more nuanced structures in fine detail. Consensus networks can also serve to highlight and evaluate causes of conflicts between multiple trees generated during bootstrapping [44,45] as these can arise from a lack of resolution (i.e., too few sequence differences between sequences) or from the recombinant nature of sequences [46,47]. In this study, both effects seem to occur. The conflicts visible in Clades 1.2 and 1.1 arise from the fact that very few variable and phylogenetically informative sites (e.g., there are only 19 within the East African/Indian clade) support some of these clades. Therefore, the process of bootstrapping did not arrive at a single supported tree structure, but rather produced multiple different trees. The conflicts observed in Clade 2.5 are, however, clearly the result of the recombinant nature of the sequences. These sequences carry the signal of both parental/ancestral sequences resulting in failure to assign a sequence reliably to either of these and effectively resulting in the pattern we see, whereby Clade 2.5 occupies a position between Clades 1.1 and 1.2.

Based on the obtained information from the phylogeny based on WGS and the consensus network analysis (Figure 3), the spatio-temporal spread of LSDV described hereafter and visualized in Figure 2A–C can be suggested. For decades after its first discovery in 1929 in what is now Zambia, LSDV has circulated only in Africa and did not disperse to other areas. 

The exact origin and dispersal trajectory cannot be recovered, but we find patterns both in the phylogeny and the decomposed network indicating two dispersals out of Africa (Clades 1.2.2, 1.2.1.1 and 1.2.1.2 see Figure 2A,B). There are very few WGS available from Africa, especially from Central and Western Africa, which creates uncertainty with regard to the exact origins of the outbreaks into Eurasia in the last decades. From 1989 onwards, LSDV spread outside its native range of Africa [48], and it has been found since the 1990s in the Middle East (reviewed in [49]). It was subsequently introduced in Southeastern Europe in 2014 (the WGS of which are described in [23,50,51]) and was detected in the Russian republic of Dagestan in 2015 and in Kazakhstan in 2016 [21]. The outbreak in Europe was controlled by a large-scale vaccination campaign using homologous live attenuated LSDV vaccines [52]. Russia implemented a vaccination program using a heterologous *Sheeppox virus* vaccine [53]. Following the vaccination campaign in Kazakhstan in 2016 whereby a live attenuated homologous LSDV vaccine was used [12], an outbreak of LSD was detected in Russia in the Saratov region [21]. The LSDV strain responsible for this outbreak was sequenced and shown to be different from all previously known field strains [22]. Subsequent investigations demonstrated that it most likely concerned a strain that resulted from multiple recombination events between a Neethling and a KSGP vaccine strain which occurred during the vaccine production process [12,20]. The vaccine turned out to contain multiple recombinants which were released in the field during the vaccination campaign in Kazakhstan. Some of these recombinant strains (Clusters 2.1, 2.2, 2.3 and 2.4 as defined by [54]) were only reported during some specific outbreaks and were not reported again on a later occasion. All currently detected recombinant field strains belong to Cluster 2.5. These strains were first detected in Western China in (2019) [55,56] and afterwards spread rapidly in East and Southeast Asia. These dispersals may have occurred on two separate occasions as indicated by the phylogeny (Figure 1B) and the decomposed network (Figure 2C). They were reported in Southern China in 2019–2020 [57]; subsequently, the strain was detected in Taiwan [58], Thailand [59], Vietnam [23,60], and Mongolia [61]. The most recent reports indicate that this strain in the meantime also spread to Indonesia where it appeared on several islands (Zainuddin presentation at the FAO LSDV meeting in March 2023). While the recombinant 2.5 strain spread rapidly in Southeast Asia, a different strain appeared in South Asia (India, Bangladesh) in 2019. The LSDV strains characterized during that outbreak in East India and Bangladesh seem closely related to Clade 1.2 Kenyan and derived KSGP strains from Eastern Africa, thereby confirming previous reports based on sequencing of shorter genome regions [62,63,64,65,66]. Again, based on sequencing of short genomic regions, these strains have now also been detected in Myanmar [67]. This outbreak, which affects the Indian subcontinent, may constitute a third dispersal of LSDV out of Africa, after the spread of the Eurasian subclade (1.2.1.2) and the appearance of recombinant Clade 2.5. The origin of these East Africa-like 1.2.2 strains in India remains unclear. Several hypotheses can be put forward, e.g., import of LSDV-infected cattle somewhere in the region or the spread of the KSGP vaccine strain (which is known to be poorly attenuated [13,50]) after its use against *Sheep*- or *Goatpox* in the region, but these remain highly hypothetical and it might turn out to be impossible to trace the true origin of this outbreak. After the finalization of our dataset for this manuscript, new sequences from strains isolated during LSDV outbreaks in the western part of India in 2022 have been reported [68]. These also seem to belong to 1.2. strains. A preliminary analysis of these sequences with the WGS presented in this paper place them in the Eurasian Clade (1.2.1.2) and are therefore different from the sequences of the 2019 outbreaks in India (clade 1.2.2). This suggests that additional LSDV incursions from other regions have occurred and emphasize that a continuous monitoring of circulating strains is warranted to closely monitor the situation in that region.

When considering the putative spatio-temporal spread of LSDV described above, it should be kept in mind that lack of sequence variation and lack of sequences throughout different regions (especially Africa) and throughout time precludes deeper and more sophisticated analyses at this time. If historical samples of LSDV-infected animals from the 20th century and from Africa are available, an effort to obtain a WGS from these should be undertaken.

### 3.2. Single-Gene Analysis Results in Recombinant Field Strain Phylogenetic Instability

In the second part of the analysis of available LSDV gene sequences, we evaluated whether phylogenetic placement based on regularly used single gene regions like rpo30, GPCR, DNA_Pol, P32 and rpo132 resulted in a similar phylogenetic placement as observed above when WGS are used. In general, the single-gene phylogenies all reflect the overall structure of the phylogeny bases on the WGS (Figure 1B and Figure 4A–E), but fewer supported clades are recovered in all cases. Three clades are supported in the rpo30-based tree (Figure 4B); four clades in the GPCR-based tree (Figure 4A); five clades in the DNA-polymerase-based tree (Figure 4C); two clades in the P32-based tree (Figure 4D) and six clades in the RPO132-based tree (Figure 4E). Clades 1.1, 1.2 and 2.5 are not always supported in each individual gene-based tree. For example, Clade 1.2 is not supported in the RPO30 and P32 gene tree and it is polyphyletic in the gene tree based on the DNA polymerase gene. Clade 1.1 is supported in the RPO30 and RPO132 gene trees and unsupported in the P32- and GPCR-based gene trees; Clade 1.1 also clustered in a clade with four recombinant strains (MH646674, OM530217, MT992618 and OL542833) in the DNA-polymerase-gene-based tree but the placement itself was not supported. Clade 2.5 forms a clade in the GPCR-, RPO30- and DNA-polymerase-based gene trees. Clade 2.5 forms a supported clade with other recombinant strains in the RPO132 and P32 trees. Importantly, the positions of the field recombinant strains belonging to Clades 2.1 to 2.4 are not stable as they ‘jump’ from clade to clade throughout the different single-gene trees (see Figure 4A–E). The Cluster 2.1 field recombinant sequences (MH646741 and OM530217) cluster in Clade 1.1 for the GPCR and RPO132 genes. They are basal to a clade that contains both Cluster 1.1 and Cluster 2.5 for the P32-based phylogeny. The field recombinant Strain 2.2 (MT134042) is assigned to Clade 1.1 in the GPCR gene tree with exactly the same genotype while it is assigned to Clade 1.2 when using RPO30 and DNA polymerase genes, and it is basal in a clade with the other recombinants and classical wild-type Clade 1.1 strains when using P32 and RPO132. Field recombinant Cluster 2.3 strain (MT992618) is a sister to Cluster 1.2 for the GPCR gene, while it shares a sequence with Strain 2.1 in the RPO30 phylogeny. Finally, field recombinant Cluster 2.4 (OL542833) is placed as a sister to Cluster 2.5 in the P32 gene tree, but it appears more closely related to Cluster 1.1 in the GPCR-, DNA-polymerase- and RPO132-based trees. In the DNA-polymerase-based gene tree, Strain 2.4 groups with Cluster 1.1. In the P32-based gene tree, it groups with Clusters 2.5, 2.1. and 1.1.

The results described above for the single-gene-based phylogenies show that it is not always possible to assign a sequence reliably to any of the subclades defined in the WGS phylogeny (Figure 1B and Figure 4); therefore, it results in uncertainty regarding the possible origin of an infection. Even worse is the fact that recombinant strains belonging to Clades 2.1 to 2.4 can be assigned to a wrong clade altogether even when using more than one gene (see Figure 4A–E). The genes assign a sample to one of the major clades correctly (classical field Strains 1.1 and 1.2 and recombinant field Strains 2.5). Therefore, using a WGS-based phylogeny is the only way to place new recombinants or variants with certainty. 

### 3.3. Genomic Substitution Hotspots in LSDV WGS Datasets

Finally, it was evaluated whether specific genomic regions were under higher evolutionary pressure than others, and whether this was different between classical (Clades 1.1 and 1.2) and recombinant (Clade 2.5) field strains. We found nine regions with elevated numbers of substitutions (>the average π-value + three times the standard deviation) (Figure 5). Two regions, LSDV008-009 and LSDV143-146, exhibited rates higher than 10 times the average π. LSDV008 encodes for a putative soluble interferon gamma receptor. Gene LSDV009 encodes a putative alpha aminitin-sensitive protein. Genes ‘LSDV 143–146′ encode for kelch-like and ankyrin repeat proteins. Six of these regions have lower, but still elevated π-values (>average π + 3 times the SD, see Figure 5) and they seem to perform other functions. The importance of these is, however, hard to determine with certainty given the function of some (e.g., LSDV132) is unknown or that the hypothesized functions are derived from Orthopox homologues or even orthologues (Table 1) [16,17,69,70,71,72,73,74]. One region, LSDV076, only shows elevated substitutions for the <2017 dataset, and it is not variable in the recombinant Clade 2.5 which carries the genotype from 1.2. LSDV076. The rarer recombinants from Cluster 2.2 (MT992618) and Cluster 2.1 (only OM530217), respectively, carry a genotype from Clade 1.1, whereas all the others carry the genotype from Clade 1.2. The LSDV076 gene encodes for a viral late transcription factor (VLTF) homologous to VLTF-4 from the Vaccinia virus.

The observation that genes encoding for a soluble interferon gamma receptor, an alpha aminitin-sensitive protein, and ankyrin repeat and kelch-like proteins are amongst the most variable genes in LSDV indicate that the virus is evolving its immune evasion capabilities. The homologues of these genes, which have been studied in greater depth in Orthopox viruses and in other pox genera, are involved in immunosuppression although the exact mechanism remains unclear, especially in LSDV [16]. They seem to have an effect on the NF-κB-coordinated anti-viral response. Some ANK repeat domains also have an F-box independent effect on viral host range [77]. Ankyrin repeats are involved in binding interferon-induced proteins, thereby misleading the immune system and delaying an immune response [76]. Kelch-like proteins interfere with the ability of the ubiquitin system to degrade proteins, thereby further hampering the induction of an adequate immune response [78]. The genomic location of these most variable genes is in the terminal regions of the genome (both 5′ and 3′) as is the case in other Poxviridae [79]. The importance of these genes in immune modulation is further underscored by the observation that they are often changed/rendered non-functional in *Capripox* vaccine-based genomes [17].

## 4. Recommendations and Remarks

Based on the results and discussion described above, we would like to argue in favor of monitoring of emerging LSDV strains by WGS. The use of single or few markers carries a real risk of misassigning a sequence to a subclade and could thereby lead to incorrect assumptions with regard to introduction routes of the virus in specific regions or on modes of transmission used by the virus. It has also been reported, for example, that recombinant strains (Clade 2.5) not only spread via vector-borne transmission, but are more efficient than classical field strains in spreading via direct and indirect contact between animals [3,6,21,22]. Whether this was indeed a distinguishing feature of the recombinant strains or simply went undetected in the classical field strains should be further investigated, and studies should be undertaken to determine which changes in the genome made this possible. Furthermore, the naïve pool of animals in Asia provides fertile grounds for further evolution and establishment of the virus as a permanent threat to livestock holders. To be able to keep an eye out for dangerous new variants, it is important to have as much genomic data available as possible. Current technologies allow to obtain sequences and high-quality assemblies in a relatively short time. While we advocate for the use of WGS for phylogeny reconstruction and identification of precise strains, we realize this is not always possible (more time-consuming, more expensive, more specialized equipment and analytical approach required). If WGS are not used and evolutionary study is required, we suggest to use multiple genes and to choose regions which are variable as these are most likely to allow for a reliable phylogenetic placement. An example of this can be found in [62], in which the authors used the R22P LSDV homologue (LSDV134). Using a higher number of genes will also allow to determine conflicts in phylogenetic placement and thus highlight potential recombinants [80].

The fact that there are currently no less than three outbreaks occurring (corresponding to Clades 1.2.1.2, 1.2.2 and 2.5) will cause the geographical ranges of the involved strains to start to overlap. This is of particular concern for strains from Cluster 2.5 and the recently discovered Clade 1.2 strains from the Indian subcontinent [60,63,65] which can be foreseen to overlap in the near future; so far, they do not yet overlap, because there is always a delay between detection and actual dispersal. That being said, strains from Clades 1.1 and 1.2 seem to have co-circulated in Africa for a long time and no naturally occurring recombinants of LSDV have been detected up till now. 

*Capripox* viruses are considered to be restricted in their host range and LSDV is considered to be restricted to bovines [81]. However, the virus has been occasionally encountered in other (non-bovine) species before, seemingly as the result of a natural transmission, e.g., in Springboks [82], giraffes, [83] and recently in camels [84]. The impact on and the role of wildlife in the expanding geographical range of LSDV remains understudied, and its epidemiological importance is unclear. Asia has a large number of wild bovines and other ruminants which might become natural reservoirs, since these species are naïve and potentially susceptible to LSD. Given the uncertainties about the future evolution and dispersal of the virus in wildlife, we advocate for vigorous monitoring in these animals, detailed geolocation of each incidence and generating a WGS whenever possible. Poxviruses are able to acquire or lose genes as well as evolve to adapt to new hosts in various ways [27,85,86], none of which can be excluded to occur in LSDV.

## Figures and Tables

**Figure 1 viruses-15-01471-f001:**
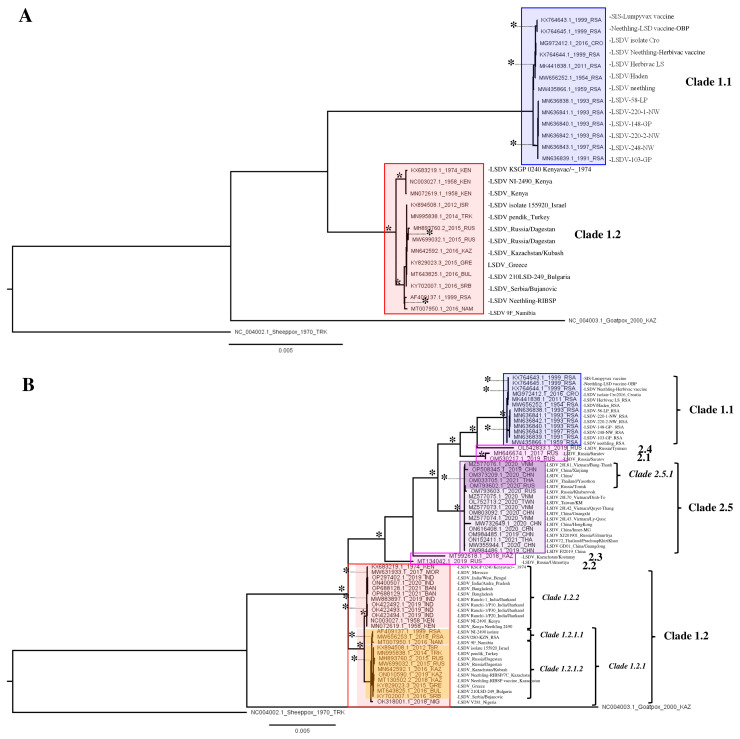
(**A**,**B**) LSDV WGS-based phylogeny for the <2017 and the >2017 dataset. Nodes (clades and subclades) supported by significant values (SH/pp/BS-s >75/>97/75) are marked with *****. The scale bare indicates substitutions per site.

**Figure 2 viruses-15-01471-f002:**
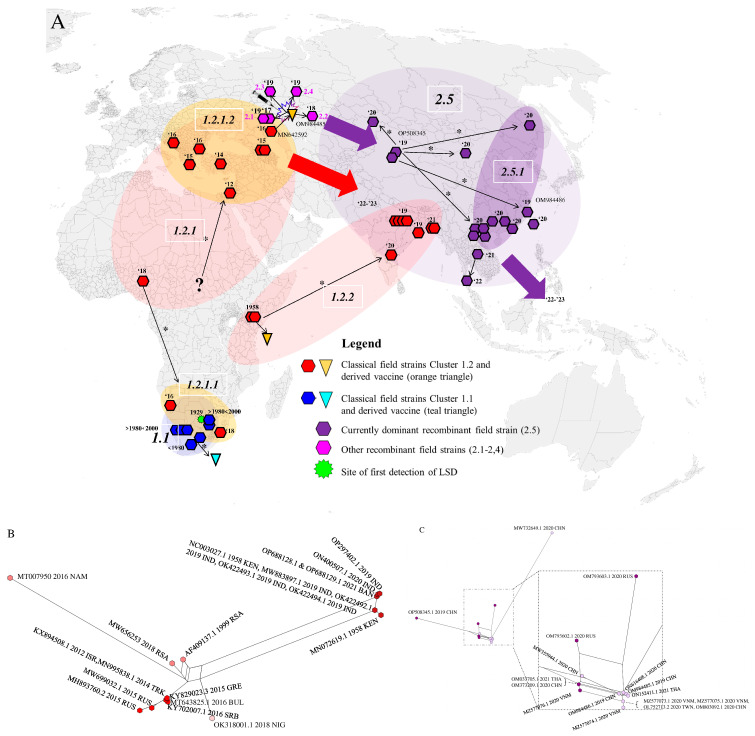
(**A**) LSDV distribution and dispersal across the world based on WGS. Major strain clades are indicated: 1.1 (blue), 1.2 (red) and 2.5 (purple) across the world. The splash symbol indicates the first occurrence of the recombinant strain. The inverted triangles indicate vaccine sequences and the respective ancestral sequences from which they were derived. The thin arrows labelled with ***** indicate relationships between accession supported by the WGS–based phylogenies. The circles/ovals indicate clades. The large arrows indicate the subsequent direction of dispersal for which the exact trajectory is unknown or not yet included in this paper. The question mark indicates uncertainty about the exact origin which led to a new outbreak. Clade labels are added in the white outlined boxes. (**B**,**C**) Networks of the sequence-based datasets for Clades 1.2 and 2.5 decomposed into splits. Here, the 2.1 and 2.5 Subclades are indicated with different colors for different clusters.

**Figure 3 viruses-15-01471-f003:**
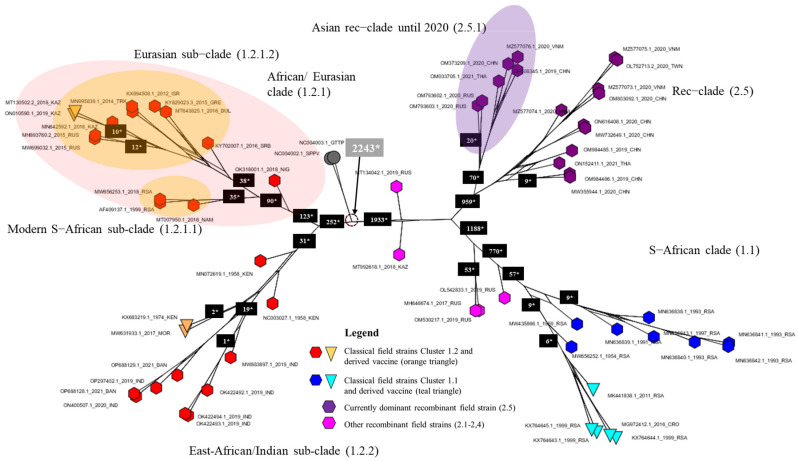
Consensus network of 10,000 trees obtained from bootstrapping for the >2017 dataset. The network is based on a count of Edge–Weights with the threshold set at 10% for displaying conflicts. The numbers on the branches indicate the number of variable positions within the group.

**Figure 4 viruses-15-01471-f004:**
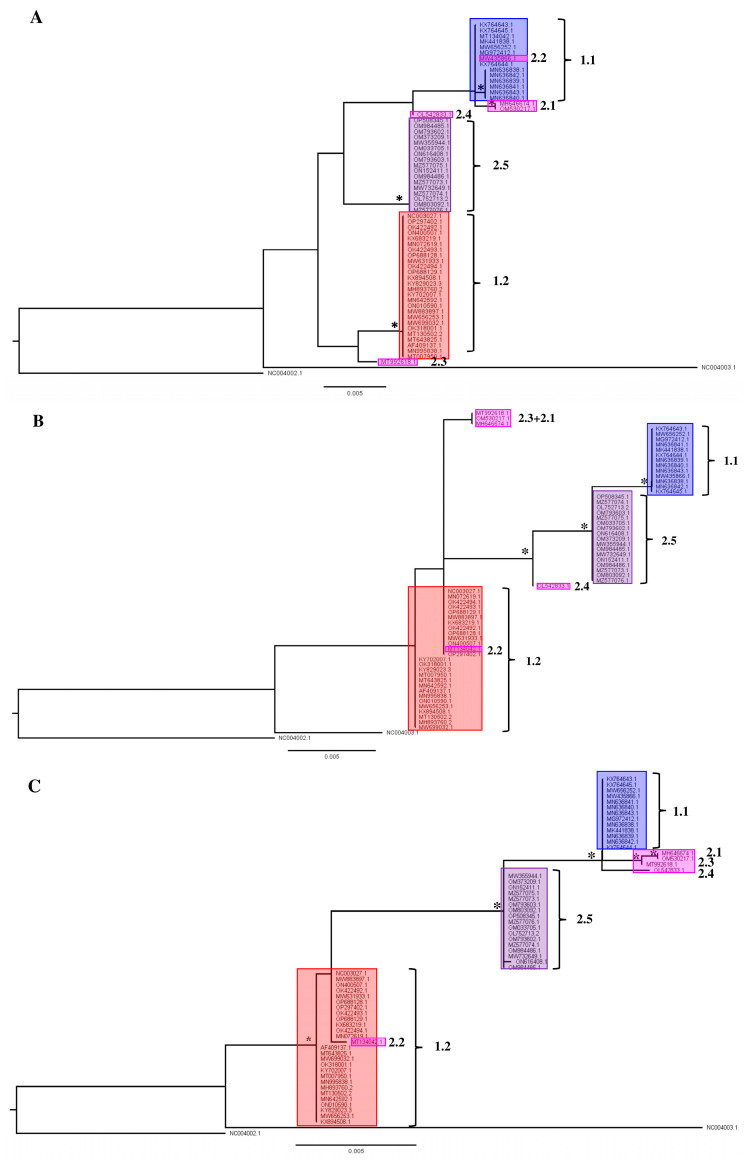
Single gene trees for (**A**) GPCR; (**B**) RNA-dependent RNA polymerase subunit 30 (RPO30); (**C**) viral DNA polymerase (DNA_Pol); (**D**) P32 gene and (**E**) RNA-dependent RNA polymerase subunit 132 (RPO132 Nodes (clades and subclades) supported by significant values (SH/pp/BS-s >75/>97/75) are marked with *****. The scale bare indicates substitutions per site.

**Figure 5 viruses-15-01471-f005:**
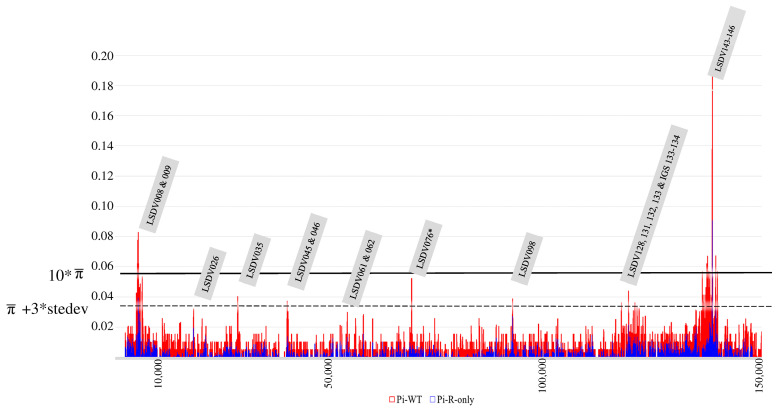
Combined π-plot of wild-type LSDV WGS (red) and recombinant LSDV WGS (blue). The genomic regions displaying peaks with >average π- + 3 times standard deviation are indicated in the grey labels. Labels with * are only affected in wild-type strains. Identification is based on the annotation of the sequence belonging to the LSDV reference strain NC_003027. The 10 * average π-value (based on the >2017 dataset) is indicated by a solid line. The average π-value + 3 times standard deviation is indicated in the figure by a dashed line. The x-axis denotes the genomic position; the y-axis denotes the π-values.

**Table 1 viruses-15-01471-t001:** Putative protein functions, mainly derived from *Vaccinia* and *Variola*.

Gene	Putative Function	References
LSDV008	A putative soluble interferon gamma receptor.	[16]
LSDV009	A putative alpha aminitin-sensitive protein.	[16]
LSDV026	Thought to be unique to LSDV in *Capripox*, acts as potential rho signaling inhibitor which may explain the formation of nodules as the ‘rho GTPases.	[75]
LSDV045	Involved in replication and recombination.	[16]
LSDV046	Involved in membrane formation.	[16]
LSDV067	Is a putative host range protein, similar to VACCP-C7L, which is suggested to interact with SAMD9. SAMD9 is involved in cell death signaling and is an innate antiviral host factor.	[16,18,76]
LSDV061	Encodes a membrane precursor of immature virions and virus factories.	[16]
LSDV062	Involved in early transcription, otherwise unknown function.	[69,70]
LSDV098	Is a protein that also encodes for the early transcription factor 82 kDa subunit. This is part of the early transcription machinery.	[69,70]
LSDV128	Membrane regulation.	[16]
LSDV132	Unknown.	[16]
LSDV133	DNA ligase.	[16]
LSDV134	Is homologous to the *Variola virus* B22R-like protein.	[73,74]
LSDV076	Late transcription factor whose *Vaccinia* homologue H5R is thought to be required both for inclusion of virosomes into crescents and for maturation of immature virions into mature virions.	[71,72]
LSDV143-146	Are kelch-like and ankyrin repeat proteins.	[16]

## Data Availability

Sequence alignments and results from phylogenetic analyses are available on request.

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
