# Peer review of "Lumpy Skin Disease Virus Genome Sequence Analysis: Putative Spatio-Temporal Epidemiology, Single Gene versus Whole Genome Phylogeny and Genomic Evolution"

_viruses, 2023, doi:10.3390/v15071471_

Round 1

Reviewer 1 Report

The manuscript well written and potentially has a big interest for research community.

Anyway, I have some minor comments:

In Introduction section statement about KSGP vaccine strain need to be explained

The LSDV KSGP strain based vaccine was originally created for the vaccina- 41 tion of sheep and goats against Sheep- and Goatpox, [13], respectively, as it was a long 42 time believed to be a sheep-goat pox strain [13,14,15].

Looks like an explanation of importance of this fact for the Introduction is missing.

All figures are hard to read because of low resolution of the images, also some mistypes need to be correct (vaccin for example). 

English looks fine for me

Author Response

Comments and Suggestions for Authors

The manuscript well written and potentially has a big interest for research community.

Anyway, I have some minor comments:

  • In Introduction section statement about KSGP vaccine strain need to be explained

The LSDV KSGP strain based vaccine was originally created for the vaccina- 41 tion of sheep and goats against Sheep- and Goatpox, [13], respectively, as it was a long 42 time believed to be a sheep-goat pox strain [13,14,15].

Looks like an explanation of importance of this fact for the Introduction is missing.

  • Reply: We have amended the sentence a bit, but to us it is more a ‘matter of fact’, there is no particular weight. It simply describes another group of vaccines. 2 paragraphs down the importance of this fact is highlighted by the fact that the faulty vaccine used in Kazachstan was composed of a mixture of strains.

  • All figures are hard to read because of low resolution of the images, also some mistypes need to be correct (vaccin for example). 
  • Reply: the figures were uploaded to ‘viruses’ in high resolution as well as included in the draft. The textual errors/inconsistencies were changed. If even higher resolution is needed the journal can contact me and I’ll try to provide them.

  • Comments on the Quality of English Language

               English looks fine for me

  • Reply: nevertheless we edited the text and improved the English and interpunction here and there.

Reviewer 2 Report

Lumpy skin disease virus (LSDV) is a bovine poxvirus that causes nodules on the skin, mucous membranes and organs.  Although some infected animals remain subclinical, they apparently can still transmit the virus, undoubtedly contributing to the morbidity rate that can reach as high as 45%.  Classical LSDV strains have been routinely divided into two major clades called 1.1 and 1.2 that, although different genetically, show no clinical differences.  However, it has become increasingly clear that this division into two clades grossly underestimates the complexity of genomic diversity among LSDV isolates. 

This manuscript details an exceptionally well-designed and exceedingly convincing comparison of the accuracy of single gene-based vs whole genome-based phylogenies.  The primary conclusion of the study is that single gene-based studies can be very simplistic and misleading, resulting in erroneous conclusions as to the routes by which the virus spreads to new regions of the world, which, in turn, compromises our ability to control that spread.  Although single gene-based sequencing is obviously significantly easier, cheaper and faster, whole genome sequencing yields a much more detailed phylogenetic and spatiotemporal resolution with respect to the spread of the virus.  In addition, the data reveal that the most highly variable regions of the genome are located in the 5’ and 3’ flanking regions that predominantly include genes responsible for immune evasion.  Based on these findings, the authors make a solid case for the use of whole gene sequencing (or at least sequencing of multiple variable genes) to construct phylogenetic trees, resulting in a more accurate reflection of the route by which the virus spreads.  Given the variability of the flanking genes and their importance to immune evasion, these regions of the genome are the obvious choice for phylogenetic analysis, if whole genome sequencing is not feasible.  In addition, since LSDV has occasionally been found in some nonbovine species, as well as wild bovines, the authors argue for monitoring these reservoirs to more closely monitor the appearance of new isolates of the virus.

This is an exceptional manuscript that puts forth a well-justified and convincing argument for the use of whole genome sequencing (WGS) , despite its cost and time-consuming nature, in phylogenetic studies of LSDV.  There can be no doubt, based on the findings presented here, that the implementation of WGS would be far more insightful and accurate in providing a true picture of LSDV phylogeny.  Indeed, the manuscript provides a roadmap for a more in depth understanding of LSDV epidemiology, which, in turn, will undoubtedly lead to better control of its spread.

Two minor points:

1)    Use of the English language and grammar is rather poor and requires considerable editorial attention;

2)    Line 22, it is not clear what the authors mean by the use of the word “misleading”.

English language usage can be improved.

Author Response

Two minor points:

  1. Use of the English language and grammar is rather poor and requires considerable editorial attention;
  • Reply: we have reviewed and revised the entire text in detail, see the accompanying document

2) Line 22, it is not clear what the authors mean by the use of the word “misleading”.

  • Reply: This sentence was truncated. We have changed it.

Comments on the Quality of English Language

English language usage can be improved.

Reply: we went through the document and made numerous edits. Please see the accompanying document for details.